# Experimental Study of In-Process Heat Treatment on the Mechanical Properties of 3D Printed Thermoplastic Polymer PLA

**DOI:** 10.3390/polym15102367

**Published:** 2023-05-18

**Authors:** Ioan Tamașag, Irina Beșliu-Băncescu, Traian-Lucian Severin, Constantin Dulucheanu, Delia-Aurora Cerlincă

**Affiliations:** Faculty of Mechanical Engineering, Automotive and Robotics, Stefan cel Mare University, 720229 Suceava, Romania; severin.traian@usm.ro (T.-L.S.); dulu@usm.ro (C.D.); delia@usm.ro (D.-A.C.)

**Keywords:** additive manufacturing, PLA, annealing, surface quality, mechanical proprieties

## Abstract

The scientific literature regarding additive manufacturing, mainly the material extrusion method, suggests that the mechanical characteristics of the parts obtained by this technology depend on a number of the input factors specific to the printing process, such as printing temperature, printing trajectory, layer height, etc., and also on the post-process operations for parts, which, unfortunately, requires supplementary setups, equipment, and multiple steps that raise the overall costs. Therefore, this paper aims to investigate the influence of the printing direction, the thickness of the deposited material layer, and the temperature of the previously deposited material layer on the part tensile strength, hardness by means of Shore D and Martens hardness, and surface finish by using an in-process annealing method. A Taguchi L9 DOE plan was developed for this purpose, where the test specimens, with dimensions according to ISO 527-2 type B, were analysed. The results showed that the presented in-process treatment method is possible and could lead to sustainable and cost-effective manufacturing processes. The varied input factors influenced all the studied parameters. Tensile strength tended to increase, up to 12.5%, when the in-process heat treatment was applied, showed a positive linear variation with nozzle diameter, and presented considerable variations with the printing direction. Shore D and Martens hardness had similar variations, and it could be observed that by applying the mentioned in-process heat treatment, the overall values tended to decrease. Printing direction had a negligible impact on the additively manufactured parts’ hardness. At the same time, the nozzle diameter presented considerable variations, up to 36% for Martens hardness and 4% for Shore D, when higher diameter nozzles were used. The ANOVA analysis highlighted that the statistically significant factors were the nozzle diameter for the part’s hardness and the printing direction for the tensile strength.

## 1. Introduction

Recently, additive manufacturing technology has been gaining momentum both in industry, with applications in the automotive, medical, aerospace, etc. industries [1], and in academia, due to the possibility of manufacturing parts with complex surfaces [2] at low costs, especially when using FDM (fused deposition modelling) [3].

In the FDM process, also known as the MEX (material extrusion) process [4], one of the most widely used materials, at the expense of petroleum-based polymers [5,6], is polylactic acid (PLA), a polymer that has a semi-crystalline structure, is biodegradable, and has good physical properties. Since the manufacturing process results in parts with low mechanical and thermal properties [6,7,8,9], the literature also presents different methods to improve these properties by reinforcing the polymer with other materials such as carbon fibre [10], glass fibre [11], and natural fibre [12], and post-processing methods, such as heat treatment [13,14,15] or chemical treatment [16,17], which involve additional steps and costs, thus also increasing the manufacturing time. Regarding the application of heat treatment on parts made of PLA by the FDM method, several research studies have been carried out [14,18,19,20,21,22], and it has been proven that the mechanical properties of the tested specimens improved drastically when the obtained parts were subjected to heat treatment post-processing. The results obtained by the authors showed that the mechanical properties are enhanced by applying a heat treatment at an optimal temperature between the polymer’s glass transition temperature and the cold crystallisation temperature [20], in the range of 80–120 °C. The literature published on these issues [14] highlights that by applying heat treatment, the material’s crystallinity level increases, which leads to an increase in mechanical properties, by up to 49%, and thermal resistance, consequently raising the glass transition point. In addition, in the case of parts made of PLA polymers, the influence of heat treatment at a temperature in the range of 65–95 °C leads to an improvement in the mechanical properties by 11–17% and in the crystallinity level by about 25% [19].

Other research [23] addresses the problem of interlayer adhesion. When using a pre-deposition heating method such as near-layer laser heating to increase the interlayer interface temperature above the critical interpenetrating diffusion temperature, the interlayer bond strength is enhanced. Results obtained by the authors showed an increase of up to 50% in the mechanical properties; the fracture behaviour of these interlayer interfaces becomes ductile, and plastic deformation is observed.

Regarding the surface hardness of additively manufactured parts, the literature presents several investigations for different types of materials used in the MEX process [24,25,26,27], including Martens-type micro-hardness [28,29,30]. Several research studies have been carried out on the influence of heat treatment on the hardness of PLA additively manufactured parts [31,32]. Following the experimental determinations in [31], the results obtained by the authors showed that by applying heat treatment, the tensile strength increased considerably, by up to 35%, while the hardness of the parts subjected to heat treatment decreased, with the values fluctuating depending on the printing direction of the specimens. Furthermore, the parts did not show plastic deformation when applying a heat treatment at a temperature in the range of 55–80 °C. Still, at a temperature of 95 °C, the parts showed a degree of dimensional deviations and were not analysed further. However, regardless of the printing direction, applying a heat treatment with temperature values in the above-mentioned range increases the ultimate tensile strength (UTS). The same trend was also observed in [32], where, by applying heat treatment in the range 70 °C−90 °C, authors observed a considerable improvement in UTS values. However, the values for surface hardness showed an increased variability, which requires further studies on the influence of heat treatment on hardness, as present in this paper.

Furthermore, in the research domain, the tendency to replace post-processing with in-processing technologies was observed. The speciality literature offers a series of research studies based on in-process applications meant to improve the mechanical properties, surface quality, or even colourisation of the parts [33,34,35]. However, related to the influence of in-process heat treatment, the literature provides limited information, especially on Martens hardness, which represents a research opportunity and provides this study the novelty of this research.

In this paper, the possibility of applying heat treatment with a hot air jet, at temperatures of 80 °C, locally, throughout the actual printing process is explored, reducing working time, manufacturing space, and the effects on the adhesion between the deposited layers. In order to solve these objectives, a Taguchi DOE L9 design was developed, which allowed the study of the influence of temperature, printing direction, and nozzle size on the tensile strength, roughness, and surface micro-hardness of the printed parts. The tests were carried out in accordance with the ISO 527 standard: Plastics—Determination of tensile properties, Parts 1 and 2 [36,37].

In Section 2, the experimental design and equipment used to achieve the proposed objectives are presented in detail, and in Section 3, the results are analysed and discussed. The experimental values obtained and included in the experimental design were analysed using Minitab software, which allowed for the plotting of variation graphs of the mean effects and Pareto graphs representing the degree of influence of each factor. At the same time, by applying the Taguchi experimental design, it was possible to apply ANOVA-type (analysis of variance) analyses to determine the statistically significant factors. The experimental results revealed that when using the in-process local heat treatment method, the values for UTS increased, and the values for hardness and surface roughness decreased. However, as observed in the literature [26], the values obtained vary depending on the printing direction and nozzle size.

## 2. Experimental Setup

To achieve the proposed objectives, a research plan shown in Figure 1 was created, which graphically illustrates the steps of this article. The ISO 527 type 1B specimens were printed according to the description in Section 2.1. They were then tested in terms of tensile strength, hardness, micro-hardness, and surface roughness with the equipment described in Section 2.2. The values obtained after the testing made in accordance with the Taguchi L9 DOE plan are analysed and discussed in Section 3.

The proposed Taguchi design of experiments (DOE) is shown in Table 1. In order to study the influence of the above-mentioned factors, three levels of variation were considered for the three factors (nozzle diameter, part temperature, and printing direction), noted with 1, 2 and 3 in Table 1 and corresponding with the values presented in Table 2.

### 2.1. Sample Preparation and Materials

The specimens made according to ISO 527-2 type B [36] have the geometry shown in Figure 2a. They were fabricated in all three directions (Figure 2b) using a Creality 3D printer, type CR6-Se; a SUNLU filament dryer, where the filament was dried before and during use at 40 °C; and a Wagner Furno 750 hot air gun, with the possibility of adjusting the temperature by 10 °C, in the range 50–630 °C (Figure 3). Furthermore, in the printing process, the Z seam position was selected to be placed in the corners of the parts to remove the influence of layer height shifting on the UTS of the specimens, as seen in the literature [13,38].

The tested material is PLA (polylactic acid) (produced by eSUN—Shenzhen Esun Industrial Co., Ltd., based in Shenzhen, China), a biodegradable and bioactive thermoplastic polymer that is commonly used in additive manufacturing due to its ease of use, low toxicity, and availability. The tested material’s chemical composition and properties according to the technical and material safety data sheets [39,40] are shown in Table 3. The printing parameters that were kept constant have the values shown in Table 4.

A FLIR X6540sc thermal imaging camera was used to study the temperature of the printing area. It was found that the temperature of the printing area shows significant variations during the printing process, with considerable decreases when new material is deposited (Figure 4).

In order to eliminate this problem, the deposition process was assisted by a hot air heating system using a Wagner blower, which maintains a constant temperature (Figure 5) throughout the deposition process (80 °C).

### 2.2. Testing Equipment

To determine the tensile strength, a GUNT testing machine was used (Figure 6a), assisted by a computer equipped with software for data processing. The values obtained experimentally were analysed, and variation graphs were made.

The surface roughness study was carried out with a Mahr CWM 100 confocal microscope and interferometer, without contact, and with NC positioning of the samples in the central region of the specimen (Figure 7). The obtained surface topography was analysed using the related MountainsLab 8.1 software, where it was possible to obtain the surface texture parameter (Sq). This parameter is an (ISO 25178) equivalent to the standard deviation of heights, which uses a predefined 0.8 mm robust Gaussian L-filter. Furthermore, the surface roughness measurements were made in a 2 × 1.5 mm^2^ area, and both directions of the roughness were taken into consideration.

The hardness of the tested parts was considered from multiple points of view. Firstly, the microhardness of test specimens was investigated and evaluated using the Shimadzu DUH-211S microhardness tester (Figure 8 to obtain the Martens hardness (Figure 9), as well as using a digital Shore D durometer to obtain the Shore hardness.

Secondly, the tests were carried out according to ISO 868:2003 [41], where the indentation position and specimen size parameters were repeated (Figure 10a) and the Shore D tests were also repeated at the edge of the specimens (Figure 10b), in the area where the part wall is created and can be considered 100% infill. The measuring tests were repeated five times, and the average value was considered.

In this study, the surface on the build platform was not taken into consideration, given that the part was printed on a tempered glass build platform and, therefore, the surface roughness is much lower than that of the rest of the part surfaces. Even more, on that specific surface, the hardness is higher. Usually, the first layers that coincide with that surface have different printing conditions, of which the most relevant are lower layer height and lower printing speeds. Therefore, the testing was made on the surfaces that differed from the build platform, and the indentation point was at least 9 mm farther from that specific surface. 

## 3. Results and Discussions

Following the experimental determinations, the values obtained have been included in Table 5, which coincides with the Taguchi L9 plan presented in Section 2, Table 1.

### 3.1. Tensile Strength

Obtained values for tensile strength ranged between 12.5 MPa and 57.5 MPa, which is also observed in other research studies [20,24,42]. Compared with the values offered by the manufacturer (Table 3), it can be seen that the results for the tensile strength from this study are lower. However, this phenomenon can be explained by the manufacturer providing the tensile strength for parts with 100% infill.

The experimental results obtained for the tensile strength allowed for drawing of dependence curves for the mean effects of the three factors considered (Figure 11). It can be seen that the tensile strength of the specimen increases substantially with increasing nozzle diameter (20.56% for 0.6 mm nozzle and 44.13% for 0.8 mm nozzle), decreases by 2.5% with increasing build platform temperature and increases by 12.5% if the temperature is kept constant at 80 °C during the printing process.

By analysing the values obtained and the graphical representation in the above graph, it was possible to draw up Table 6, where the symbols ↑ and ↓ represent growth and decline, respectively.

The results obtained confirm the conclusions presented in the works [19,43] concerning the tensile strength and the influence of the nozzle size. An increase of 20.5% in the tensile strength can be observed when using the 0.6 mm nozzle and 19.5% when using the 0.8 mm nozzle. These values are explained by the reduction in the number of layers used for printing the walls, but also in the interlayer contact dimension. Other explanations for this phenomenon may be due to the increased adhesion between the layers due to the density changes of the test specimens, proven in the literature [43,44], where the authors concluded that by increasing the nozzle size, the density of the specimens also increases. Another explanation may be the fact that the wider the nozzle diameter, the longer the layers stay at a maintenance temperature that allows the material to settle better due to thermal convection [34].

Regarding the influence of the printing direction on the tensile strength, the experimental tests showed an increase of 14.11% in the case of the Y-direction and 61.35% in the case of the Z-direction, variations similar to those found in the literature [18].

However, compared to the X-direction, higher values for the tensile strength were obtained when parts were printed in the Y-direction. This may also be due to changes in the cross-section of the specimens. Due to the positioning and orientation of the lines of the layers, the tensile strength varies considerably. Figure 12 shows the settings used for printing in the X and Y directions.

The lines of the outer walls are different from one printing direction to another, which influences the area of occurrence of the bottleneck where the material breaks and the morphology of the apparent section of the specimen, and, therefore, the tensile strength. In the present case, the high results of tensile strength for parts printed in the Y-direction are due to the fact that the total volume of the upper and lower surfaces of the specimen is kept constant by depositing material in the form of continuous, straight lines at 0–90°.

By applying the experimental design presented in subchapter 2.1, it was possible to employ an ANOVA analysis (Table 7), from which resulted the level of statistical significance (Figure 13).

The results obtained from the ANOVA analysis showed an almost insignificant variation in the tensile strength when only the temperature of the printing platform was used, whereas when the heating method described in Section 2.1 was used, the values showed an improvement of up to 12.5% in tensile strength. Given this, it can be concluded that by applying local heating, the tensile strength increases, confirming other research studies from the literature related to the application of heat treatment [31,32,45]. However, the values obtained for the tensile strength have a smaller variation than those obtained by other researchers between parts without annealing and parts with annealing. This may be due to the duration of the treatment application. In most of the research studies, the experiments involved increased treatment application times, whereas in this case, the exposure time of the parts to local heat treatment depends on the part size and printing speed. In this case, the duration of local heat treatment on the parts coincides with the printing duration, calculated in the slicing software Cura 5.2.2, shown in Table 5. Of the factors studied, the ANOVA analysis revealed that the statistically significant factors influencing the tensile strength variation were printing direction with 65.2% influence, followed by nozzle diameter with 24.68% influence, and finally, temperature with 10.12% influence.

### 3.2. Hardness

The surface hardness results gave similar values regardless of indentation position. For this reason, only the results of tests performed according to ISO 969:2003 were considered. Similarly, the variations in the type of test show the same trends. The graphs in Figure 14 and Figure 15 represent the variation in the mean effects of Shore D hardness and Martens, respectively.

Following the variation graphs shown above, it can be considered that the values obtained for hardness are opposite to those for tensile strength. Comparing the results obtained with those found in the literature, it is found that they fall within the range of 79–84 HD, values also obtained by other researchers [26,31], raising the degree of confidence in the accuracy of the values obtained in this study.

The level of variation and direction of hardness trends for both types of tests are presented in Table 8.

Contrary to the trends observed for the tensile tests, it can be seen that surface hardness decreases with increasing nozzle diameter by up to 3.97% for Shore D hardness and 36.76% for Martens hardness when nozzles larger than 0.5 mm in diameter were used. Research in the field [43] has shown that in MEX fabrication, even when 100% infill was used, air voids occur between the deposited layers due to improper adhesion. It was observed that by increasing the nozzle diameter, the apparent density of the specimens changes, and significant voids appear in the internal structure of the specimens with sizes up to 254 µm when using 0.8 mm diameter nozzles. Moreover, in some cases, the material shows air pores due to nozzle wear or gas evacuation produced by filament moisture [34], leading to morphological changes in the material and, consequently, to the size and homogeneity of the specimen’s apparent section (Figure 16).

In this sense, it can be concluded that the variation for hardness obtained with the nozzle diameter is due to the thickness of the deposited material layer’s lines, which implicitly leads to a larger volume of air pores between the layers, thus resulting in lower values for the hardness of the material studied.

By using the local heating method, the hardness trend is decreasing, an aspect also observed by other researchers [31] who used conventional post-processing annealing methods. The recorded values showed up to 1.44% variation for Shore hardness and 8.64% for Martens hardness when 80 °C temperature was used. This trend, in addition to the morphological changes of the semi-crystalline structure material, may also be due to the reduction in the internal stresses of the heat-treated material, as concluded by other researchers [15,18].

The hardness values as a function of printing direction have slight variations. However, their trend has also been observed in the literature [45], where it was deduced that the highest hardness was obtained when the horizontal printing direction was used, and the lowest values were recorded when the parts were printed in the Z-direction.

The ANOVA analysis revealed that, of all the factors studied for both types of hardness, the statistically significant factor (*p*-value < 0.05) was nozzle diameter (Table 9 for Shore hardness and Table 10 for Martens hardness).

However, considering the Pareto charts in Figure 17 and Figure 18, it can be seen that the local heating application also has a high degree of influence. The significance levels for both types of hardness have been listed in Table 10.

From Table 11, it can be seen that temperature influences Martens hardness by about 20% and Shore hardness by 20%, showing that the application of this type of local heating has an influence on the mechanical characteristics of the resulting parts. Moreover, the variations presented in this paper in both the hardness of the parts and the tensile strength have the same trends as those found in the literature, raising the confidence in the application of the in-process local heat treatment method.

### 3.3. Surface Roughness

Surface roughness measurements were made on the surfaces coinciding with the layer height, in this case, as shown in Figure 19, on the red surfaces. However, the values for the surfaces parallel to the build platform were also compared as a function of nozzle diameter.

As specified in Section 2 for hardness, the surface roughness tests also did not take into account the surface on the printer platform. However, Table 12 shows comparative results between a surface on the build platform and an opposing surface. Moreover, the table also shows a comparison between a simple tempered glass build platform and a build platform covered by microporous coating. It can be seen that due to the different manufacturing conditions for the first layers and the use of tempered glass as the build platform, the values showed a big difference, with the lowest roughness being obtained for the surface on the build platform.

The results from Table 12 show a major difference between the surface roughness on the build platform and the surface roughness of the surfaces opposite to the build platform. In this case, the simple tempered glass build platform corresponded to the best surface roughness values (Sq = 2.63 µm, compared to Sq = 5.10 µm obtained for the coated build platform). The microporous coating used for the build platform was a carborundum (SiC) produced by Creality 3D, which is specific for the 3D printer used in the 3D experiments. However, the surface opposite from the build platform presented up to 38× higher surface roughness values compared to those obtained for the two types of build platforms.

On the basis of the experimental results for surface roughness, it was possible to generate the mean effect plots (Figure 20) and significance plots of the factors studied, resulting from an ANOVA analysis (Table 13, Figure 21).

In the case of parts printed in any direction, and regardless of the heat treatment, due to the fact that the layer height was kept constant, the results for surface roughness as a function of nozzle diameter showed very small variations, up to 2.81%, with the values falling within the range of 48–49.5 µm. However, the presented analysis shows an increasing trend for surface roughness values with increasing nozzle diameter, an aspect also observed in other studies [46,47].

The analysis highlighted that the in-process heat treatment had the most impact on the resulting surface roughness. A small variation of 6.25% was obtained when only the build platform temperature was used, followed by a considerable improvement in the surface roughness by 63.17% when the in-process heat treatment was used, compared with no heat. Additionally, an ANOVA analysis showed that the statistically significant factor, on surface roughness, with 79.53% significance, was temperature.

Printing direction shows neglectable variation for the X and Y directions, while an 8.19% increase in surface roughness was observed for parts manufactured in the Z direction. This aspect may be explained by the parts’ dimensions and the used 3D printer type. The used 3D printer was a bed slinger type, which uses the build platform to move in the Y-direction, influencing the stability of thin and tall parts.

Table 14 presents the overall variation percentage and the degree of influence resulting from the analysis. An ANOVA analysis shows the degree of influence of all the studied factors. In this study, the statistically significant factor (*p*-value < 0.5) was the part temperature with 79.53% influence, followed by printing direction with 14.92% influence, and finally, the nozzle size with 5.55% influence.

Another important aspect was observed when the results for surface roughness were compared with the values obtained for the flat surfaces, perpendicular with the nozzle (Figure 19). Figure 22 shows the results for surface roughness as a function of nozzle diameter, for the surfaces of the part parallel and opposite to the build platform.

From Figure 22, it can be seen that, contrary to the results for surface roughness for surfaces that coincide with layer height (Figure 20), the values tend to decrease with increasing nozzle size. This aspect is justified by the fact that with increasing nozzle diameter, the interlayer adhesions are fewer and depend on the extrusion width. With increasing nozzle diameter, extrusion width also increases, requiring fewer layers to print a surface.

## 4. Conclusions

The analysis of the results obtained in this paper, related to the factors of influence considered, shows that the direction of printing and the nozzle diameter have the most significant influence on the tensile strength. In this case, given the input factors used, the temperature of the part does not have a major influence on the tensile strength, but it is not negligible. With the results obtained, it can be considered that the application of heat treatment during the process is possible. Keeping the temperature constant has influences on the tensile strength up to the critical melting temperature (the temperature at which the material plasticises massively), after which the deposition process cannot be continued.

The experimental values and trends are in agreement with the results obtained by other researchers in the literature [18,31,32,47,48]. This convergence of results from multiple sources significantly bolsters the confidence level associated with the outcomes presented in this paper, further confirms the possibility of applying in-process annealing during additive manufacturing, and also strengthens the reliability and significance of the multiple research studies from the literature.

For this study, the presented in-process heat treatment was applied on small parts with flat surfaces, with the heat being blown in one direction. Following the previous conclusion and the results from this study, the in-process heat treatment methodology will be optimised in the future, which offers the opportunity for more studies regarding the in-process heat treatment impact on larger parts with complex shapes.

The in-process heat treatment helps with the adhesion of the layers, resulting in an improvement in tensile strength by about 12.5% and an improved roughness by 12%, which demonstrates that the application of in-process heat treatment is feasible and can provide opportunities for further research by optimising the process. Regarding the hardness of the material, the results obtained confirmed the results of other research [28] and showed that by applying in-process heat treatment, due to morphological changes of the semi-crystalline material, the hardness decreases.

Nozzle diameter positively influences tensile strength, leading to an increase of about 44% when using a 0.8 mm diameter nozzle. At the same time, air voids in the material, as demonstrated in the literature [43], lead to a significant decrease in material hardness, with nozzle diameter being the factor with the highest statistical significance (*p*-value < 0.05).

Surface texture is an important factor for part functionality. In MEX processes the surface roughness is determined by the specific way in which melted material flows through the extruding nozzle and is deposited in the part as layers. The extruded material’s proprieties are also strongly influenced by the printing temperature and cooling speed. The in-process heat treatment technique reheats the deposited material layers and slows the cooling of the new material layers that are being deposited. The experimental data showed that, for parts with higher surface quality demands, if in-process heat treatment is considered, the previous layer temperature should be one of the main parameters that needs careful selection. The surface roughness Sq parameter improves by 63.17% when in-process heat treatment is used and is approximately constant over the entire printed part. For the specific printing conditions considered in this study, the nozzle diameter and the printing direction parameter did not exhibit a significant influence on the surface roughness.

The printing direction greatly impacted the tensile strength, where the lowest values were obtained when the parts were printed in the Z-direction and the highest when printed in the Y-direction. Upon hardness, the printing direction had a low influence.

## Figures and Tables

**Figure 1 polymers-15-02367-f001:**
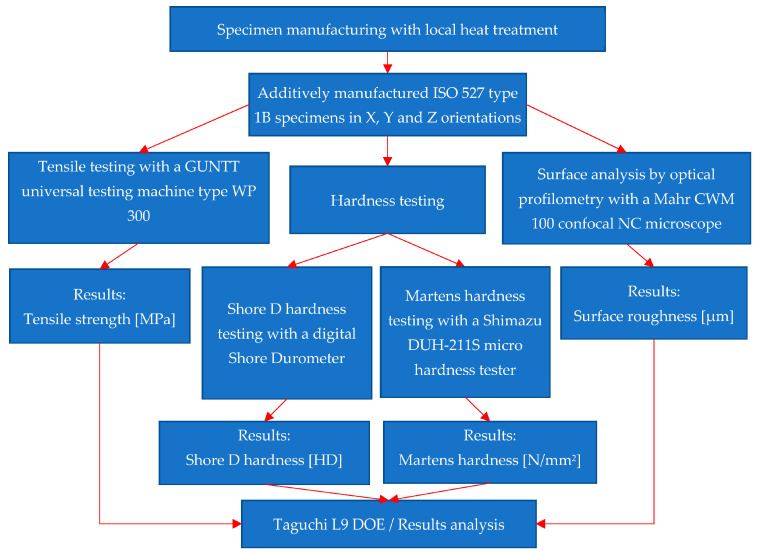
The experimental program.

**Figure 2 polymers-15-02367-f002:**
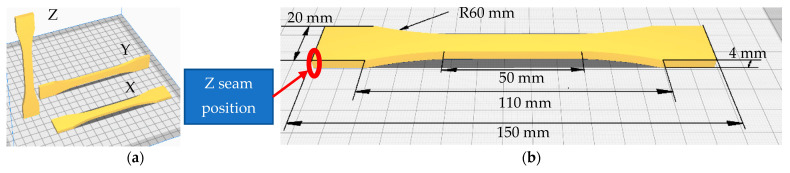
ISO 527 type 1B specimens: (**a**) printing directions; (**b**) specimen dimensions.

**Figure 3 polymers-15-02367-f003:**
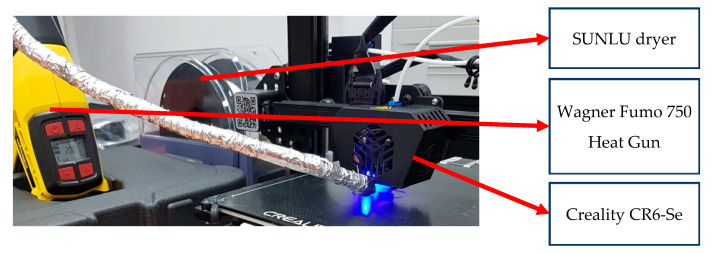
3D printer assembly.

**Figure 4 polymers-15-02367-f004:**
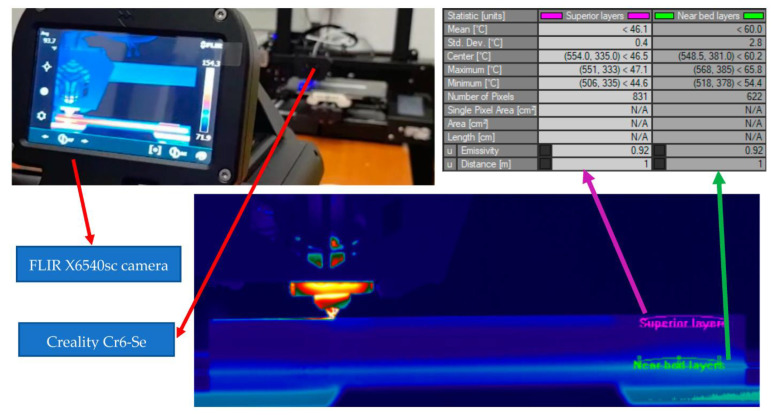
Temperature difference between printed layers.

**Figure 5 polymers-15-02367-f005:**
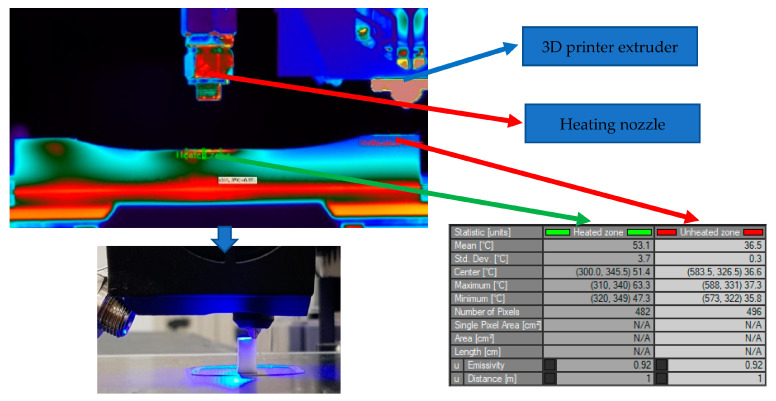
Local heating of the part.

**Figure 6 polymers-15-02367-f006:**
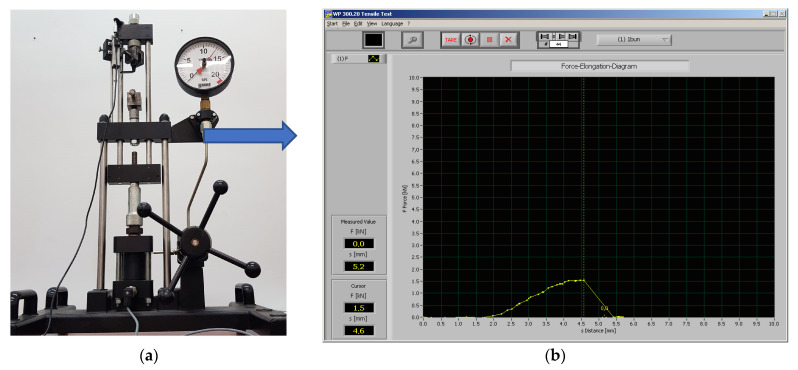
GUNT universal testing unit: (**a**) testing machine; (**b**) obtained tensile strength result image.

**Figure 7 polymers-15-02367-f007:**
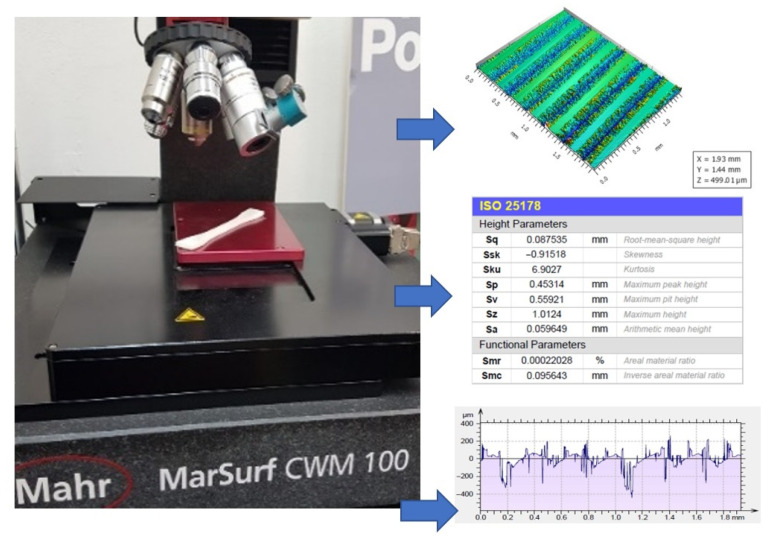
Mahr CWM 100 confocal microscope.

**Figure 8 polymers-15-02367-f008:**
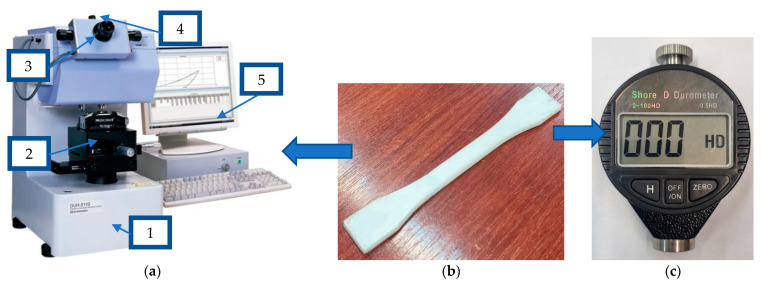
Hardness testing: (**a**) Shimadzu DUH-211S microhardness tester (1), sample manual positioning system (2), footprint optical viewing system (3), image pickup video system CCD (4), hardness measurement and footprint inspection software (5); (**b**) test specimen; (**c**) Shore D durometer.

**Figure 9 polymers-15-02367-f009:**
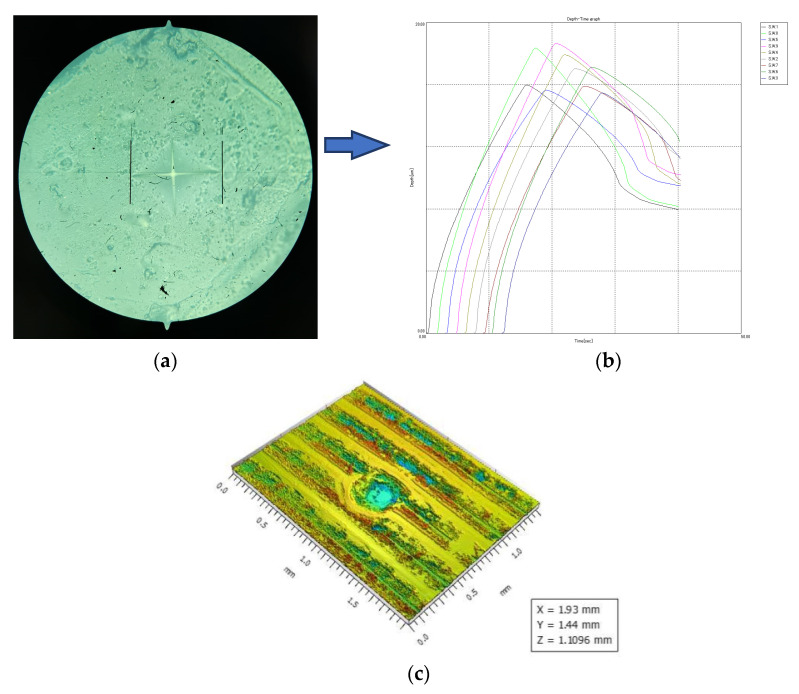
Hardness testing results: (**a**) optical image; (**b**) resulted graph; (**c**) indentation point.

**Figure 10 polymers-15-02367-f010:**
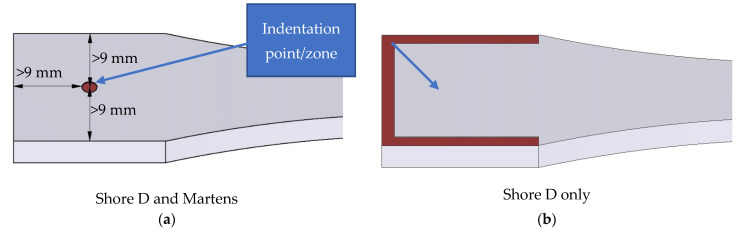
Indentation position: (**a**) in accordance with ISO 969:2003; (**b**) on walls.

**Figure 11 polymers-15-02367-f011:**
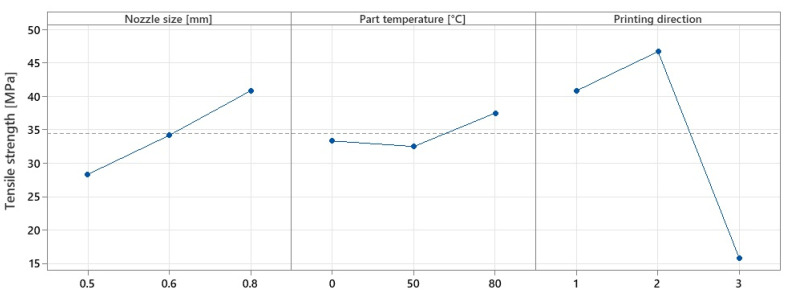
Variation in mean effects for tensile strength.

**Figure 12 polymers-15-02367-f012:**
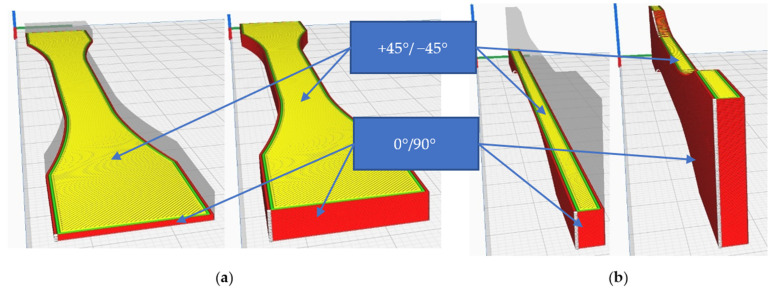
Test specimens: (**a**) flat specimen, printed in the X-direction; (**b**) specimen printed in Y-direction.

**Figure 13 polymers-15-02367-f013:**
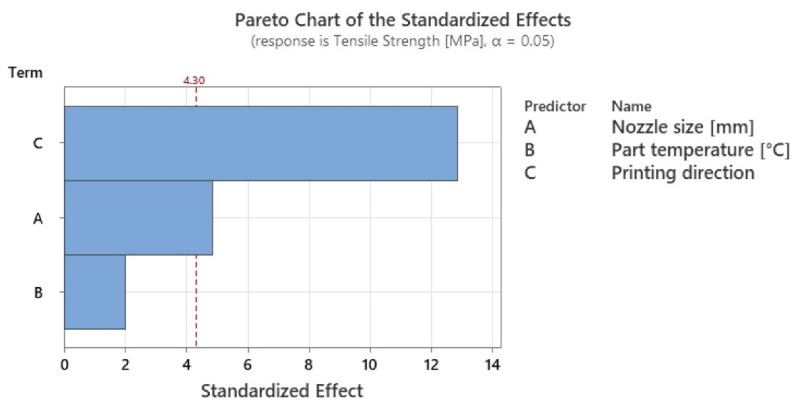
Graphical illustration of the degree of influence of studied factors on tensile strength.

**Figure 14 polymers-15-02367-f014:**
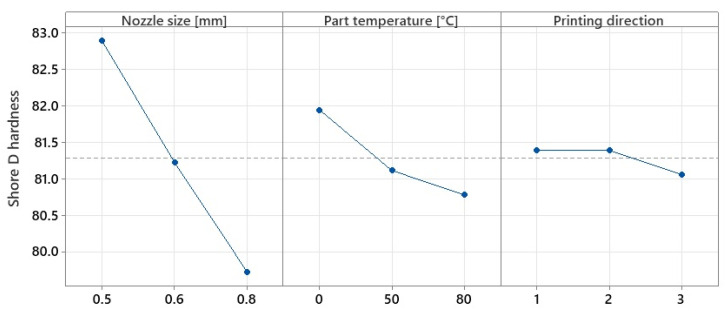
Variation in mean effects for Shore D hardness.

**Figure 15 polymers-15-02367-f015:**
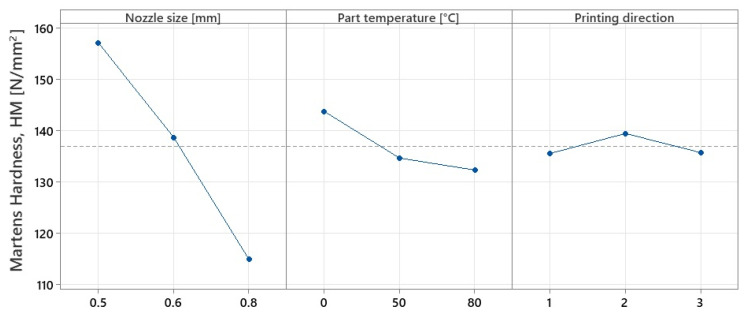
Variation in mean effects for Martens hardness.

**Figure 16 polymers-15-02367-f016:**
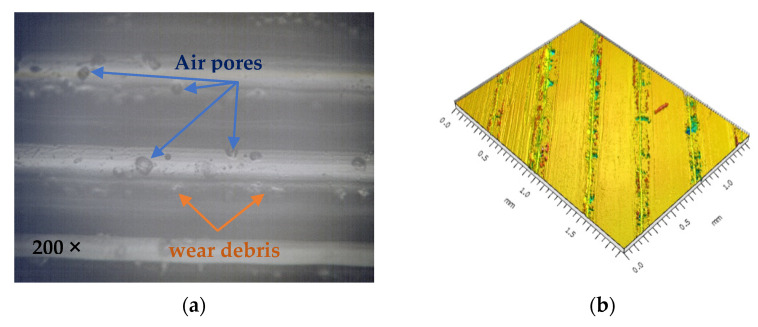
Surface analysis: (**a**) microscopical view; (**b**) surface topography [34].

**Figure 17 polymers-15-02367-f017:**
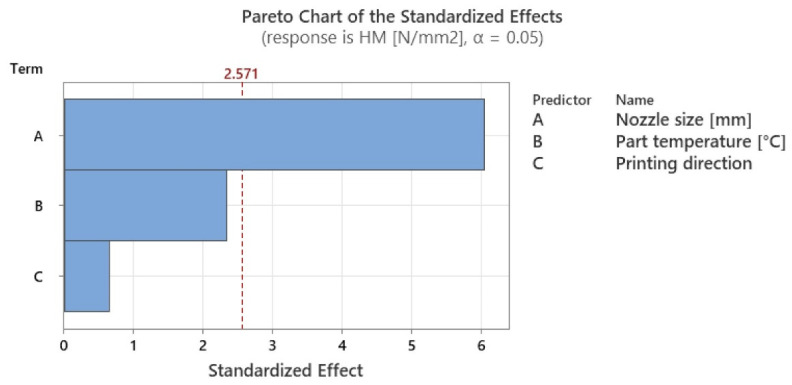
Graphical illustration of the degree of influence of studied factors on Shore D hardness.

**Figure 18 polymers-15-02367-f018:**
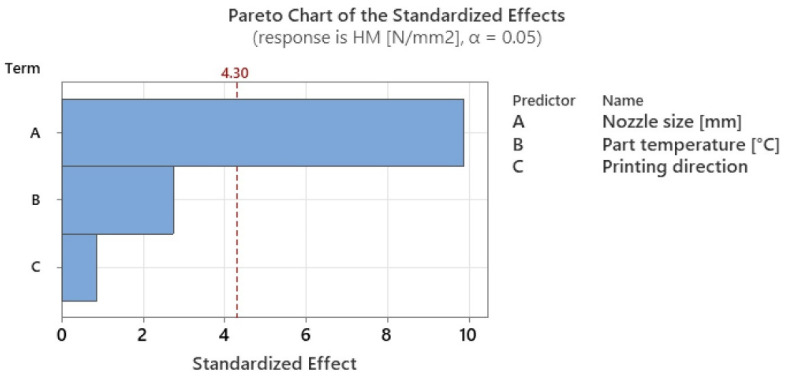
Graphical illustration of the degree of influence of studied factors on Martens hardness.

**Figure 19 polymers-15-02367-f019:**
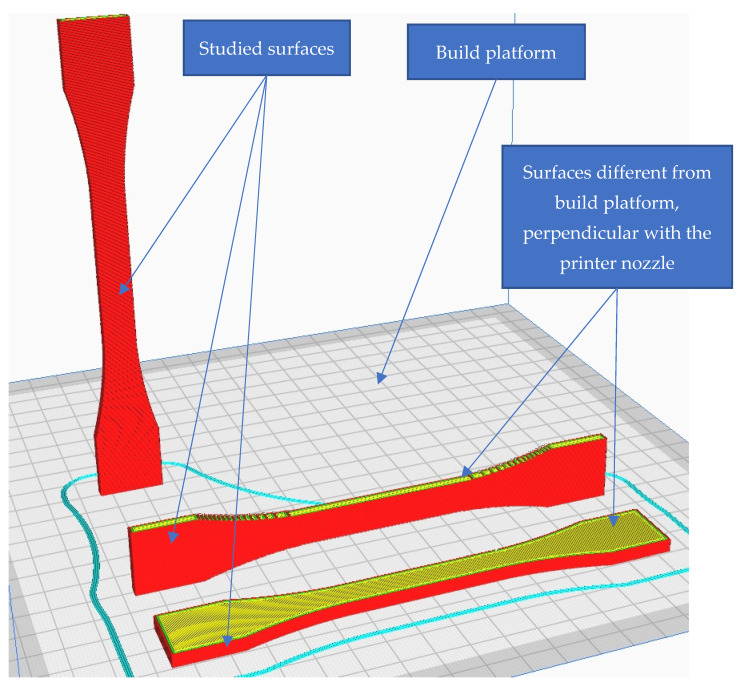
Graphical illustration of the measured zones for surface roughness.

**Figure 20 polymers-15-02367-f020:**
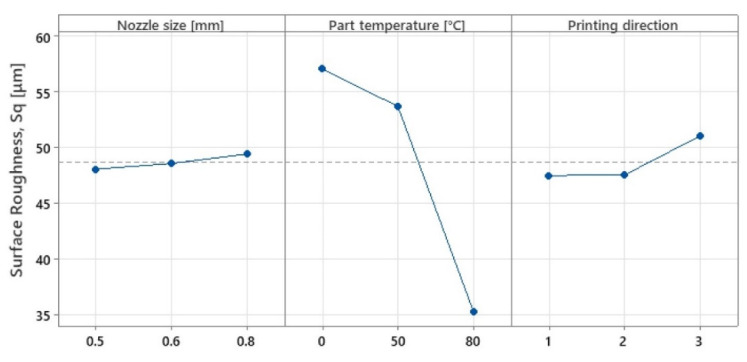
Variation in mean effects for surface roughness.

**Figure 21 polymers-15-02367-f021:**
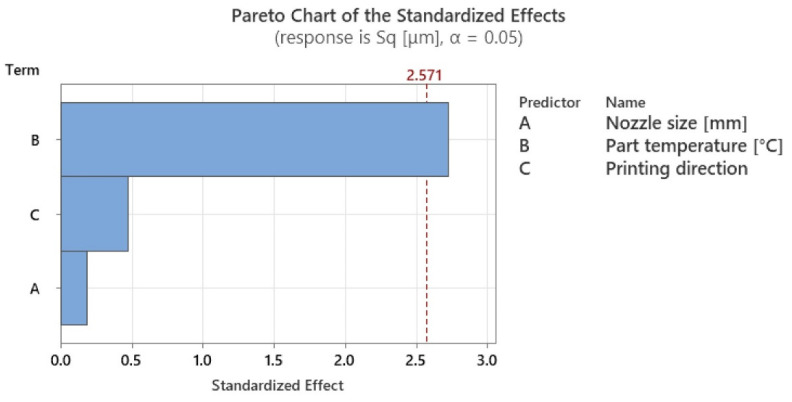
Graphical illustration of the degree of influence of studied factors on surface roughness.

**Figure 22 polymers-15-02367-f022:**
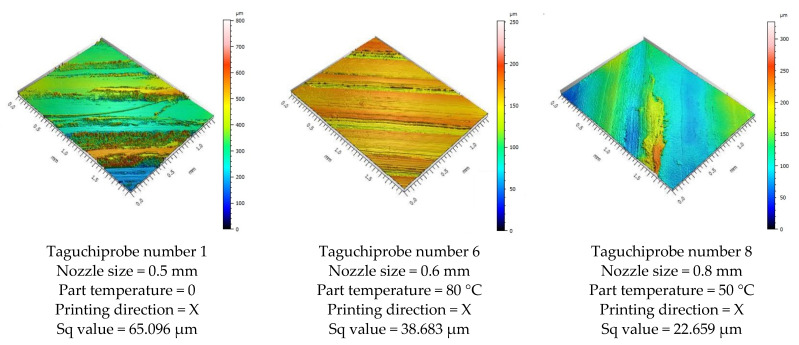
Nozzle diameter influence on the top surface roughness of the tested specimens.

**Table 1 polymers-15-02367-t001:** Taguchi L27 DOE.

Test Number	Nozzle Size	Heat Temperature	Printing Direction
1	1	1	1
2		2	2
3		3	3
4	2	1	2
5		2	3
6		3	1
7	3	1	3
8		2	1
9		3	2

**Table 2 polymers-15-02367-t002:** Levels of variation for the factors used in the Taguchi DOE design.

Levels	A	B	C
	Nozzle Size [mm]	Part Temperature [°C]	Printing Direction
1	0.5 mm	No heat	X
2	0.6 mm	Heated build platform at 50 °C	Y
3	0.8 mm	Local heating at 80 °C	Z

**Table 3 polymers-15-02367-t003:** Material properties.

Material	Density (g/cm^3^)	Heat Distortion Temp (℃, 0.45 MPa)	Melt Flow Index (g/10 min)	Tensile Strength [MPa]	Flexural Modulus [MPa]	Polylactic Acid
PLA	1.2	53	3.5	75	1915	99.8%

**Table 4 polymers-15-02367-t004:** Additive manufacturing constant parameters.

Parameter	Values	Parameter	Values
Layer height	0.3 [mm]	Print speed	50 [mm/s]
Infill	20%	Travel speed	150 [mm/s]
Infill type	Cubic	Retraction distance	6.5 [mm]
Printing temperature	215 [°C]	Cooling	ON

**Table 5 polymers-15-02367-t005:** Experimental values.

Test Number	Tensile Strength [MPa]	Shore D Hardness [HD]	Martens Hardness, HMV [N/mm^2^]	Printing Duration [min]	Surface Roughness, Sq [µm]
1	35.0	84.0	160.765	32	49.237
2	37.5	83.0	156.843	38	57.613
3	12.5	81.7	153.672	61	37.431
4	45.0	81.5	150.406	33	56.215
5	15.0	81.2	133.415	64	49.923
6	42.5	81.0	132.185	28	38.683
7	20.0	80.3	119.970	59	65.826
8	45.0	79.2	113.623	24	53.660
9	57.5	79.7	111.014	29	28.857

**Table 6 polymers-15-02367-t006:** Obtained results’ percentage variation for tensile strength.

Level	Nozzle Size		Part Temperature		Printing Direction	
	Value	Variation	Value	Variation	Value	Variation
1	0.5	-	0	-	X	-
2	0.6	↑ 20.56%	50	↓ 2.5%	Y	↑ 14.11%
3	0.8	↑ 44.13%	80	↑ 12.5%	Z	↓ 61.35%

**Table 7 polymers-15-02367-t007:** ANOVA analysis for tensile strength.

Source	DF	Adj SS	Adj MS	F-Value	*p*-Value
Regression	6	1887.50	314.583	64.71	0.015
Nozzle size	2	234.72	117.361	24.14	0.040
Part temperature	2	43.06	21.528	4.43	0.184
Printing direction	2	1609.72	804.861	165.57	0.006
Error	2	9.72	4.861		
Total	8	1897.22			

**Table 8 polymers-15-02367-t008:** The level of variation and direction of hardness trends for both types of tests.

	Level	Nozzle Size		Part Temperature		Printing Direction	
		Value	Variation	Value	Variation	Value	Variation
Shore D	1	0.5	-	0	-	X	-
	2	0.6	↓ 2.05%	50	↓ 1.03%	Y	- 0%
	3	0.8	↓ 3.97%	80	↓ 1.44%	Z	↓ 0.41%
		Value	Variation	Value	Variation	Value	Variation
Martens	1	0.5	-	0	-	X	-
	2	0.6	↓ 13.29%	50	↓ 6.75%	Y	↑ 2.88%
	3	0.8	↓ 36.76%	80	↓ 8.64%	Z	↑ 0.08%

**Table 9 polymers-15-02367-t009:** ANOVA analysis for Shore D hardness.

Source	DF	Adj SS	Adj MS	F-Value	*p*-Value
Regression	3	16.6474	5.5491	14.13	0.007
Nozzle size	1	14.3353	14.3353	36.50	0.002
Part temperature	1	2.1454	2.1454	5.46	0.067
Printing direction	1	0.1667	0.1667	0.42	0.544
Error	5	1.9637	0.3927		
Total	8	18.6111			

**Table 10 polymers-15-02367-t010:** ANOVA analysis for Martens hardness.

Source	DF	Adj SS	Adj MS	F-Value	*p*-Value
Regression	6	2936.47	489.41	35.72	0.027
Nozzle size	2	2688.79	1344.39	98.12	0.010
Part temperature	2	218.52	109.26	7.97	0.111
Printing direction	2	29.16	14.58	1.06	0.484
Error	2	27.40	13.70		
Total	8	2963.87			

**Table 11 polymers-15-02367-t011:** Obtained results’ percentage variation for hardness.

Factor	Shore D	Martens
Nozzle size	66.90%	73.35%
Temperature	25.88%	20.34%
Printing direction	7.210%	6.320%

**Table 12 polymers-15-02367-t012:** Roughness values for surfaces on the build platform and opposite.

	Surface on Simple Tempered Glass Build Platform	Surface on Coated Tempered Glass Build Platform	Surface Opposite to Build Platform
3D image	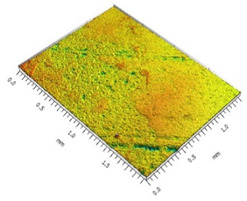	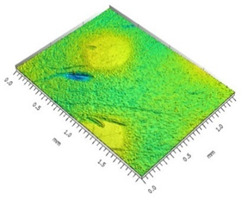	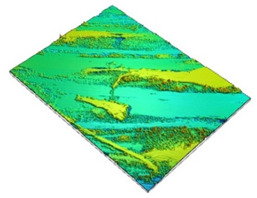
2D image	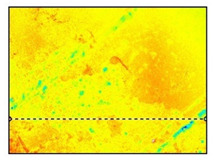	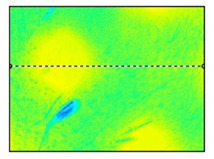	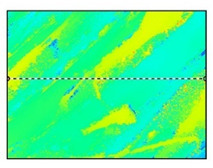
Profile	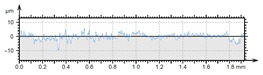	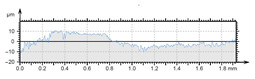	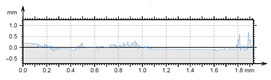
Sq	2.63 µm	5.10 µm	99.3 µm

**Table 13 polymers-15-02367-t013:** ANOVA analysis for surface roughness.

Source	DF	Adj SS	Adj MS	F-Value	*p*-Value
Regression	3	639.94	213.313	2.55	0.169
Nozzle size	1	2.80	2.796	0.03	0.862
Part temperature	1	618.42	618.416	7.40	0.042
Printing direction	1	18.73	18.727	0.22	0.656
Error	5	417.70	83.541		
Total	8	1057.64			

**Table 14 polymers-15-02367-t014:** Obtained results’ percentage variation and degree of influence for surface roughness.

Level	Nozzle Size		Part Temperature		Printing Direction	
	Value	Variation	Value	Variation	Value	Variation
1	0.5	-	0	-	X	-
2	0.6	↑ 0.37%	50	↓ 6.25%	Y	↑ 0.78%
3	0.8	↑ 2.81%	80	↓ 63.17%	Z	↑ 8.19%
Degree of variation according to the ANOVA analysis						
Percentage	5.55%		79.53%		14.92%	

## Data Availability

Some or all data, models, or code generated or used during the study are available from the corresponding author by request.

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
