# Peer review of "Experimental Study of In-Process Heat Treatment on the Mechanical Properties of 3D Printed Thermoplastic Polymer PLA"

_polymers, 2023, doi:10.3390/polym15102367_

Round 1
Reviewer 1 Report
In this paper, mechanical performances of PLA specimens (under heat treatment) manufactured by FFF process was investigated experimentally.
1. The abstract is very general, and there is a lot of irrelevant information. The abstract should be explained and showed the important aspects of work. So, this abstract in the present form is unacceptable. Please remove equipment names, filaments, etc. from the abstract
2. Bibliographic source no. 7 is not included in the text. Also, package-type citations should be better detailed (example: [11, 15-19]).
3. The current stage should be completed with references that include works on PLA filament reinforced with: carbon fibre (https://doi.org/10.3390/polym14235230), glass fibre (https://doi.org/10.3390/polym14224988) and natural fiber (https://doi.org/10.3390/polym14071301).
4. In Table 3, tensile strength and flexural strength from the manufacturer's website must be added. The authors must make a comparison between the results obtained by them and those of the manufacturer.
5. In figure 4 - the bottom left table is not understandable. And the data in that table is important.
6. Figure 8 must be redone. Captions must be added for each figure.
6. The Results and discussions chapter must be redone.
A. Tensile test
Example of a phrase that needs to be redone..... “Regarding the influence of the printing direction, the experimental tests showed an increase of 14.11% in the case of the Y direction and 61.35% in the case of the Z direction, variations similar to those found in the literature [15].” Whose growth? However, compared to the X-direction, higher values were obtained when parts were printed in the Y-direction. Whose value? Tensile strengths … probable
How many specimens were tensile tested? According to the standard and for a correct validation of the results, 5 specimens must be tested. Otherwise, the results are irrelevant. There are no graphs (Stress - Strain or Load-Displacement) related to tensile tests. For printing on Y you needed support material. Did the connection area, which also corresponds to the support area, influence the mechanical tensile properties?
B. Hardness
Did the printing position of the specimens have any influence on the microhardness? The 2 faces should be analyzed (the upper face and the lower face of the specimen - exemplification - the placement face and the top face - where the part to be printed is finished). You will notice different values for the 2 types of tests.
C. Surface roughness
The roughness according to the three diameters of the nozzle must be analyzed. The authors must add new graphs and analyzes on roughness and establish some clear conclusions that can be compared with other results in the field (since they exist).
The paper must be rewritten and checked because it contains many ambiguous sentences.
Moderate editing of English language
Author Response
Reviewer's comment no. 1. The abstract is very general, and there is a lot of irrelevant information. The abstract should be explained and showed the important aspects of work. So, this abstract in the present form is unacceptable. Please remove equipment names, filaments, etc. from the abstract.
Author’s response to the reviewer's comment: The abstract was reconsidered. See revised paper.
Reviewer's comment no. 2. Bibliographic source no. 7 is not included in the text. Also, package-type citations should be better detailed (example: [11, 15-19]).
Author’s response to the reviewer's comment: Bibliographic reference 7 was included in text and a more details were added in the introduction chapter for package-type citations.
Reviewer's comment no. 3. The current stage should be completed with references that include works on PLA filament reinforced with: carbon fibre (https://doi.org/10.3390/polym14235230), glass fibre (https://doi.org/10.3390/polym14224988) and natural fiber (https://doi.org/10.3390/polym14071301).
Author’s response to the reviewer's comment: The revised paper takes into consideration the reviewer suggestion. The specified references were included into the introduction chapter.
Reviewer's comment no. 4. In Table 3, tensile strength and flexural strength from the manufacturer's website must be added. The authors must make a comparison between the results obtained by them and those of the manufacturer.
Author’s response to the reviewer's comment. Table 3 was completed with the tensile strength values offered by the manufacturer. The results from this study were compared with the ones offered by the manufacturer and a conclusion was drown. See revised paper, lines 224-228
Reviewer's comment no. 5. In figure 4 - the bottom left table is not understandable. And the data in that table is important.
Author’s response to the reviewer's comment: Figure 4 was reconsidered. Bottom left image and the table with temperatures was resized for more clarity.
Reviewer's comment no. 6. Figure 8 must be redone. Captions must be added for each figure.
Author’s response to the reviewer's comment: Figure 8 was reconsidered, sliced and mentioned in text as figures 8 and 9. Now the Captions for all elements in the figures were added.
Reviewer's comment no. 7: The Results and discussions chapter must be redone.
- Tensile test
Example of a phrase that needs to be redone..... “Regarding the influence of the printing direction, the experimental tests showed an increase of 14.11% in the case of the Y direction and 61.35% in the case of the Z direction, variations similar to those found in the literature [15].” Whose growth? However, compared to the X-direction, higher values were obtained when parts were printed in the Y-direction. Whose value? Tensile strengths … probable
How many specimens were tensile tested? According to the standard and for a correct validation of the results, 5 specimens must be tested. Otherwise, the results are irrelevant. There are no graphs (Stress - Strain or Load-Displacement) related to tensile tests. For printing on Y you needed support material. Did the connection area, which also corresponds to the support area, influence the mechanical tensile properties?
- Hardness
Did the printing position of the specimens have any influence on the microhardness? The 2 faces should be analyzed (the upper face and the lower face of the specimen - exemplification - the placement face and the top face - where the part to be printed is finished). You will notice different values for the 2 types of tests.
- Surface roughness
The roughness according to the three diameters of the nozzle must be analyzed. The authors must add new graphs and analyzes on roughness and establish some clear conclusions that can be compared with other results in the field (since they exist).
The paper must be rewritten and checked because it contains many ambiguous sentences.
.
Author’s response to the reviewer's comment: The results and discussions section was improved with the reviewer’s suggestions as follows:
- Tensile test
The text was improved with the reviewer’s suggestions, and ambiguous phrases were clarified.
In accordance to the mentioned standard, the minimum number of 5 test samples were used for each test, and the measurements for Hardness and Surface roughness were repetead multiple times to see the repetability of the obtained results. The mean values were further considered when the tables and graphs were created in DOE analysis software Minitab.
A example graph with the tensile strength recorded by Gunt tensile machine is given in figure 6.b.
It is true that printing on Y needed supports, however, the settings for the supports in Cura were selected to be Tree support, with moderate density, to minimize the effect of supports on the surface. In the case of the specimens used for this paper, the supports removed easily and did not left any impurities on the necking zones of the specimens.
- Hardness
Yes, the reviwer is right. In this study the surface on the build platform was not taken into consideration given the fact that the part was printed on a tempered glass build platform, therefore the surface roughness is much better than the rest of the surfaces. Even more, on that specific surface, the hardness is higher. Usually, the first layers, that coincide with that surface has different printing conditions from which the most relevant are: lower layer high and smaller printing speeds. Therefore, the testings were made on the surfaces that differ from the build platform one, and the identation point was with at least 9mm farther from that specific surface or part edge. These clarifications were introduced into the revised paper. Also, a comparative example was introduced to Surface Roughness section where the aspect evidentiated by the reviewer was higlighted. See lines 206-212 for Hardness and lines 368-381 for Surface roughness.
- Surface roughness
Surface roughness subsection was completly revised. Variation graphs, Anova analysis, Surface images and comparisions were introduced into the revised paper.
Reviewer 2 Report
The paper presents, in the case of FDM (MEX) process, PLA material, research on the influence of the manufacturing direction, the thickness of the deposited material layer and the temperature of the layer previously deposited by using an in-process annealing method, on the tensile strength, hardness and surface quality of the part.
In my opinion, the article will be better appreciated by the readers if the following points will be improved:
- Do not use or specify abbreviations and round brackets in the abstract;
- In the keywords section is better to use the following order of keywords: “Additive Manufacturing; PLA; Annealing, Surface Quality, Mechanical Proprieties”.
- In the paper, instead of “3D printing” is recommended to be used the scientific term “additive manufacturing”;
- Figure 7 is not so clear, is better to be resized;
- In the Conclusion section emphasize why the research is so important because you write “The experimental values and trends are in agreement with the results obtained by other researchers in the literature….”and detail what you will propose to do in the future (in the research direction of the paper).
Author Response
Reviewer's comment no. 1: Do not use or specify abbreviations and round brackets in the abstract.
Author’s response to the reviewer's comment: The abstract was reconsidered entirely. See revised paper.
Reviewer's comment no. 2. In the keywords section is better to use the following order of keywords: “Additive Manufacturing; PLA; Annealing, Surface Quality, Mechanical Proprieties”.
Author’s response to the reviewer's comment: Thank you for the suggestion. In the revised paper the keywords were modified accordingly.
Reviewer's comment no. 3. In the paper, instead of “3D printing” is recommended to be used the scientific term “additive manufacturing”.
Author’s response to the reviewer's comment: Thank you for the suggestion. In the revised paper the terms were modified accordingly. Moreover, other terms were modified to coincide with ISO/ASTM 52900; International Standard: 2021 Additive Manufacturing—General Principles—Fundamentals and Vocabulary.
Reviewer's comment no. 4. Figure 7 is not so clear, is better to be resized.
Author’s response to the reviewer's comment: Figure 7 was reconsidered and resized. See revised paper.
Reviewer's comment no. 5. In the Conclusion section emphasize why the research is so important because you write “The experimental values and trends are in agreement with the results obtained by other researchers in the literature….”and detail what you will propose to do in the future (in the research direction of the paper).
Authors response to the reviewer's comment: The reviewer’s suggestion was taken into consideration. In the conclusions section, more details about the significance of the research and possible future studies were added.
Reviewer 3 Report
1. 3D printing is currently the most popular technology in the production of prototypes. With the development of this technology, the quality of products improves. The mechanism of the process is similar to extrusion, mainly in terms of the working tool. The authors raise a very important topic related to the method of layering and the thermal effect on the final properties of the product.
2. The introduction is well-formulated, however, the manuscript lacks a bibliography, despite the fact that the authors provide references to literature in the main text.
3. The adopted research methodology is appropriate.
4. Table 1 is not entirely clear.
5. The figures are correctly presented, the photos complement the research material.
6. Figure 12, 16, 17 needs quality improvement.
7. Concluding conclusions are correctly formulated.
Author Response
Reviewer's comment no. 1. The introduction is well-formulated, however, the manuscript lacks a bibliography, despite the fact that the authors provide references to literature in the main text.
Author’s response to the reviewer's comment. Thank you for the observation. The bibliographic reference was introduced in the text, in Introduction section.
Reviewer's comment no. 2. The adopted research methodology is appropriate.
Author’s response to the reviewer's comment. Thank you.
Reviewer's comment no. 3. Table 1 is not entirely clear.
Author’s response to the reviewer's comment. A clearer description of table 1 was introduced in the revised paper. See lines 136-139.
Reviewer's comment no. 4. The figures are correctly presented, the photos complement the research material.
Author’s response to the reviewer's comment. Thank you.
Reviewer's comment no. 5. Figure 12, 16, 17 needs quality improvement.
Author’s response to the reviewer's comment. The figures were corrected and the used font was enlarged for better clarity. In the revised paper, figures 12, 16 and 17 became 13, 17 and 18. See revised paper.
Reviewer's comment no. 6. I Concluding conclusions are correctly formulated.
Author’s response to the reviewer's comment. Thank you.
Round 2
Reviewer 1 Report
The authors significantly improved the manuscript by introducing some edifying paragraphs. The manuscript is ready for publication.
Moderate editing of English language